# Evolution of Interfacial Friction Angle and Contact Area of Polymer Pellets during the Initial Stage of Ultrasonic Plasticization

**DOI:** 10.3390/polym11122103

**Published:** 2019-12-14

**Authors:** Bingyan Jiang, Yang Zou, Guomeng Wei, Wangqing Wu

**Affiliations:** 1State Key Laboratory of High-Performance Complex Manufacturing, Central South University, Lushan South Road 932, Changsha 410083, China; jby@csu.edu.cn (B.J.); jal8276478@163.com (Y.Z.); guomengweicsu@163.com (G.W.); 2School of Mechanical and Electrical Engineering, Central South University, Lushan South Road 932, Changsha 410083, China

**Keywords:** ultrasonic plasticization, micro-injection molding, interfacial friction heating, polymer tribology

## Abstract

Interfacial friction heating is one of the leading heat generation mechanisms during the initial stage of ultrasonic plasticization of polymer pellets, which has a significant influence on the subsequent viscoelastic heating according to our previous study. The interfacial friction angle and contact area of polymer pellets are critical boundary conditions for the analysis of interfacial frictional heating of polymer pellets. However, the duration of the interfacial friction heating is extremely short in ultrasonic plasticization, and the polymer pellets are randomly distributed in the cylindrical barrel, resulting in the characterization of the distribution of the interfacial friction angle and contact area to be a challenge. In this work, the interfacial friction angle of the polymer pellets in the partially plasticized samples of polymethyl methacrylate (PMMA), polypropylene (PP), and nylon66 (PA66) were characterized by a super-high magnification lens zoom 3D microscope. The influence of trigger pressure, plasticizing pressure, ultrasonic amplitude, and vibration time on the interfacial friction angle and the contact area of the polymer pellets were studied by a single factor experiment. The results show that the compaction degree of the plasticized samples could be enhanced by increasing the level of the process parameters. With the increasing parameter level, the proportion of interfacial friction angle in the range of 0–10° and 80–90° increased, while the proportion in the range of 30–60° decreased accordingly. The proportion of the contact area of the polymer pellets was increased up to 50% of the interfacial friction area which includes the upper, lower, and side area of the cylindrical plasticized sample.

## 1. Introduction

With the advantages of low production cost, high molding accuracy, and easy to mass production, micro-injection molding has become one of the key technologies in micro/nano manufacturing and been widely used in various fields like micro-optics, microfluidic chips, and micro sensors and actuators, etc. [1]. Conventional micro-injection molding machines faced many challenges such as high energy consumption, difficulty in filling micro-cavities, and low material utilization, the negative impact is particularly obvious in mass production [2]. To address the challenges of conventional micro-injection molding, Michaeli [3] proposed an ultrasonic plasticization concept for micro-injection molding (UPMIM) technology in 2002 with the advantages of low energy consumption, good melt fluidity, and short molding time [4,5]. In UPMIM, a small amount of polymer pellets can be plasticized only by the ultrasonic vibration energy without external heating. The ultrasonic vibration energy is introduced by the sonotrode which is further used as a plunger to inject the plasticized polymer melt into the micro cavities as shown in Figure 1. To explore the potential of UPMIM technology, researchers have studied the application of UPMIM in micro-scale cavity filling [6,7], micro-molding performance [4,8,9,10,11], material blending modification [12,13,14], and high viscosity material molding [15,16,17]. It was found that the UPMIM technology is extremely effective in polymer plasticization and highly energy efficient in mold filling. Unfortunately, the UPMIM technology is subjected to a low process stability such as the wild fluctuation of the introduced ultrasonic vibration energy under the same process conditions. This could be ascribed to the plasticization process during which the interaction among the polymer pellets is highly stochastic and time dependent, leading to an extremely complicated plasticizing heat generation mechanism. 

Heat generation in polymer pellets under ultrasonic vibration is one of the critical issues due to its influence on the melting quality. Michaeli believes that the energy source of ultrasonic plasticization mainly comes from frictional heat and viscoelastic heat, similar to ultrasonic welding [18]. We studied the mechanism of frictional heat generation and viscoelastic heat generation [19,20,21,22] and found that the interfacial frictional heat determines the temperature distribution during the initial stage of ultrasonic plasticization. Interfacial friction heat generation is the main energy source when the polymer pellets are at low temperatures. When the temperature increases up to the glass transition point, the decrease of polymer elasticity leads to a decrease of frictional heat generation rate. Meanwhile, viscoelastic heat generation becomes the main plasticizing energy source. 

In order to research the mechanism of frictional heat generation during ultrasonic plasticization, we established a simplified physical model [22] referring to ultrasonic welding [23,24,25]. A polymer rod was cut off with an angle of 30° between the cross-section and the horizontal plane to study the interfacial friction heat generation under ultrasonic vibration. It was found that the simplified experimental model is inadequate for the study of the interfacial friction heat generation, but no relevant research was found. Since the state of the contact area of ultrasonic welding is certain [26], it is easy to establish a reasonable geometric model by referring to the contact state of the welding surface. The research of frictional heat generation in the ultrasonic plasticization process faced many difficulties, such as the random stack of pellets and extremely short duration of the interfacial friction heating. It is difficult to characterize the evolution of interfacial friction angle and contact area of polymer pellets during ultrasonic plasticization due to the contact state change between pellets in the plasticization process. 

The evolution and distribution of the interfacial friction angle of polymer pellets during the ultrasonic plasticization process were studied with partially plasticized samples obtained at various time intervals. The interfacial friction angle between polymer pellets was measured by a super-high magnification lens zoom 3D microscope (SMLM). The influence of trigger pressure, plasticizing pressure, ultrasonic amplitude, vibration time, and material type on the interfacial friction angle and the contact area of the polymer pellets were studied by a single factor experiment. The results can be used to establish an accurate geometric analysis model of friction heat generation. 

## 2. Experimentation

### 2.1. Materials

Overall there are three kinds of material used in this study, i.e., polymethyl methacrylate (PMMA, elliptic cylinder, TF8, Mitsubishi Chemical Holdings Group, Japan), polypropylene (PP, water drop-shaped, K1011, Formosa Chemicals & Fiber Corporation, Taiwan, China), and nylon 66 (PA66, elliptic cylinder, Zytel101NC010, DuPont Corporation, Shunde, China). Material properties are as shown in Table 1. 

### 2.2. Equipment

The experimental device used in this study is a self-developed UPMIM prototype machine which basically consists of an ultrasonic vibration system and a servo motion control system as shown in Figure 2. The ultrasonic vibration system provides polymer plasticizing energy, consists of ultrasonic power supply, transducer, and sonotrode. The frequency of the ultrasonic vibration system is 20 kHz ± 500 Hz and the amplitude adjustment range is 18–32 μm. The movement of the sonotrode is controlled by the servo motion control system. The servo motor provides the plasticizing pressure of the sonotrode. The plasticizing pressure range used in the experiment is 10–30 MPa. The cavity of the plasticizing chamber is a cylinder with a diameter of 10 mm and a height of 45 mm. The mold temperature has a significant influence on the melt fluidity [27,28] but has little effect on the plasticizing process. Therefore, the influence of the mold temperature in the plasticized process is ignored in this paper. 

### 2.3. Methodology 

#### 2.3.1. Experiment Process 

Pressure triggering and position triggering are two different methods to activate the ultrasonic power supply during plasticization. Pressure triggering refers to the control mode that the sonotrode starts to vibrate when it contacts the polymer and reaches the specified value of pressure. Position triggering refers to the control mode that the sonotrode starts to vibrate when it moves to the specified position. The polymer pellets were unplasticized before triggering the ultrasonic supply. Two experiments are designed respectively to study the influence of trigger pressure and ultrasonic parameters on the evolution of interfacial friction angle and contact area of pellets during the initial stage of ultrasonic plasticization. 

Due to the weak interface adhesion of pellets during triggering pressure experiments, Ethyl α-cyanoacrylate (ECA) was used to increase adhesion. The material was dried before the experiment. The drying temperature and time of PMMA were 80 °C and 4 h; the drying temperature and time of PA66 were 100 °C and 4 h; PP is weak in moisture absorption and does not require drying. In the experiment, 0.35 g pellets were sufficiently mixed with 0.15 mL ECA, then the mixture was placed into the plasticizing chamber. 10, 20, and 30 MPa were selected for the trigger pressure. Five pre-experiments were carried out for each parameter to make the equipment run stably; each group of parameters was repeated three times after pre-experiments. The servo motor was turned off after the sonotrode presses the polymer pellets at a specific pressure for 2 s (the motor is self-locking after turning off), the samples were taken out after curing for 30 min. The adhesion strength of PP and ECA is weak for PP as it is a non-polar polymer, then we polished PP pellets by 200 mesh sandpaper to increase the adhesion strength. 

The influences of plasticizing pressure, ultrasonic amplitude, and vibration time on the interfacial friction angle and the contact area of the polymer pellets were studied by a single factor experiment. The parameter combinations are shown in Table 2. Each group of the parameter was repeated three times after five pre-experiments. For each experiment, 0.35 g polymer pellets were used.

#### 2.3.2. Measurement

After preparing the plasticized sample, the interfacial friction angle of pellets was measured by SMLM (VHX-5000, KEYENCE, Osaka, Japan). The method, process, and result of measurement are shown in Figure 3 and Figure 4. The friction surface to be measured was selected before measuring the inter-pellets friction angle, as shown in Figure 3a. The pellets on the end or flank surface of the plasticized sample were sequentially removed by a tweezer, as shown in Figure 3b. The plasticized sample is placed on the measuring platform, and the coordinates of the highest two points of the contact surface slope are measured, as shown in Figure 3c. The friction angle between the pellets is shown in Figure 3d. The interfacial friction angle of pellets is calculated by Formula (1). θ is the friction angle (the angle between the friction surface and the horizontal plane) of the two pellets. X2 and y2 are the horizontal and vertical coordinates of the higher point in the vertical direction. X1 and y1 are the horizontal and vertical coordinates of the lower point in the vertical direction. The proportion of the interfacial friction angle of pellets is calculated by Formula (2). F is the proportion of angle distribution (e.g., 0–10°), nα is the number of contact surfaces at a specific angle, and nθ is the number of interfacial contact surfaces of the pellet. The friction angle was measured by SMLM, as shown in Figure 4a. A sample from which some of the pellets were removed is shown in Figure 4b. The friction angle measurement results are shown in Figure 4c.
Θ = arctan(x_2_ − x_1_)/(y_2_ − y_1_)(1)
F = n_α_/n_θ_*100%(2)

## 3. Results and Discussion 

### 3.1. Height of Plasticized Sample 

The influence of the trigger pressure on the height distribution of the plasticized sample was as shown in Figure 5. The theoretical compaction height can be sequenced as PP > PA66 > PMMA, which were 4.95, 3.91, and 3.75 mm, respectively. When the trigger pressure was 10 MPa, the heights of PP, PA66, and PMMA samples were 8.83, 7.93, and 7.22 mm respectively, which were much higher than the theoretical compaction height, there were many voids inside the sample. When the trigger pressure was increased to 30 MPa, the heights of PP, PA66, and PMMA samples were reduced by 6.9%, 9%, and 8.9%, respectively. In this paper, we hold that the plasticized sample was in a compacted state when the difference between the actual height and the theoretical compaction height was less than 5%; thus, the sample was uncompacted under the range of current parameters. The plasticized sample height can be sequenced as PP > PA66 > PMMA. The main reason for the height of the sample being much higher than the theoretical compaction height maybe that polymer pellets were unplasticized in triggering experiments. Relatively low sonotrode pressure leads to the tiny material deformation; the height of the sample mainly depends on the density of the material. Increasing the trigger pressure will slightly increase the compaction degree of the sample. 

The influence of the plasticizing pressure on the height distribution of the plasticized sample was as shown in Figure 6. Increasing the plasticizing pressure can increase the positive pressure and contact area of the friction pair, improve the heat generation efficiency, and increase the compaction degree of the sample. When the plasticizing pressure was 10 MPa, the sample heights of PP, PA66, and PMMA were 6.42, 5.74, and 5.51 mm, respectively. When the plasticizing pressure was increased to 30 MPa, the heights of PP, PA66, and PMMA samples were reduced by 22.56%, 33.1%, and 33.3%, respectively. All of the samples were in compacted statues while the parameter level was high. Compared with the experimental results of trigger pressure, the height of PP, PA66, and PMMA samples decreased by 43.1%, 49.8%, and 47.3%, respectively, the influence of plasticizing pressure can be sequenced as PMMA ≈ PA66 > PP. The possible reason for the above results was that the temperature of the contact area of the pellet increases rapidly under interfacial friction [19,21,22]. The pressure of sonotrode makes the softened polymer fill the gap, the contact state of inter-pellets changes from point contact to face contact meanwhile. The results show that ultrasonic vibration can significantly improve the density of plasticized samples. 

The influence of the ultrasonic amplitude on the height distribution of the plasticized sample was as shown in Figure 7. Increased ultrasonic amplitude not only increases the sliding speed between friction pairs but also increases the viscous dissipation of polymers, which increases heat generation efficiency. When the ultrasonic amplitude was 20 μm, the heights of PP, PA66, and PMMA samples were 5.92, 5.21, and 5.05 mm, respectively. When the plasticizing pressure increased to 30 μm, the height of PP, PA66, and PMMA samples were reduced by 14.8%, 21.9%, and 24.1%, respectively. All of the samples were in a compacted status when the parameter level was high. The influence of ultrasonic amplitude can be sequenced as PMMA > PA66 > PP. Compared with the plasticizing pressure, the ultrasonic amplitude has less influence on the compaction degree of samples. 

The influence of the vibration time on the height distribution of the plasticized sample was as shown in Figure 8. When the vibration time was 1 s, the heights of PP, PA66, and PMMA samples were 5.89, 4.70, and 4.56 mm, respectively. When the vibration time increased to 2 s, the heights of PP, PA66, and PMMA samples were reduced by 13.8%, 13.2%, and 17.8%, respectively. All of the samples were in a compacted status when the parameter level was high. The influence of vibration time on compaction degree can be sequenced as PMMA > PA66 > PP, which may be related to the energy required for the respective materials to reach the melting point. For 0.35 g PMMA, PP, and PA66, the energy required to rise from room temperature to melting temperature was 61.53, 104.45, and 136.01 J, respectively. The difference in energy required to reach the melting point may be the main cause of the difference in the density of the plasticized sample. It can be seen from Figure 6, Figure 7 and Figure 8 that all of the samples were in a compacted status when the ultrasonic parameter level was high, so the parameter selection range was reasonable. 

### 3.2. Interfacial Friction Angle of the Polymer Pellets

#### 3.2.1. Influence of Trigger Pressure

The interfacial friction angle of pellets can be divided into five types: 0–10° (close to horizontal), 10–30° (small tile angle), 30–60° (medium tilt angle), 60–80° (large tile angle), and 80–90° (close to vertical). When the trigger pressure range was 10–30 MPa, the interfacial friction angle of the PMMA, PP, and PA66 pellets were as shown in Figure 9a–c. The proportion of friction angles of 30–60° was relatively high, and the proportion of other angles were similar. When the trigger pressure increased from 10 to 30 MPa, the proportion of 30–60° decreased comparatively significantly (about 15%) while the angle variation of others were comparatively small. The influence of trigger pressure on the distribution of the interfacial friction angle of pellets was as shown in Figure 9d. For PMMA, PP, and PA66, when the trigger pressure increased to 30 MPa, the proportion of friction angle of 0–30° were 39.8% (+9.3%, compared with the lowest parameter level, similarly hereinafter), 33.2% (+3.9%), and 27.9% (+1.2%); The proportion of 30–60° were 34.6% (−5.1%), 33.1% (−13.4%), and 29.4% (−16.1%). The proportion of 60–90° were 32.7% (+2.9%), 28.6% (+4.5%), and 42.2% (+14.5%). With increasing the trigger pressure, the proportion of interfacial friction angle in the range of 0–30° and 60–90° was increased, while the proportion in the range of 30–60° was decreased accordingly. As can be seen from Figure 5, the trigger pressure was inversely related to the height of the sample. When the height of the sample decreases, the void between polymer pellet tends to fill up, the interfacial friction angle of pellet tends to be horizontal (0–10°) and vertical (80–90°). From the results, the proportion of interfacial friction angle of pellet in the range of 30–60° was decreased, while the proportion in the range of 0–10°and the 80–90° was increased accordingly. The influence of trigger pressure on the friction angle can be sequenced as PA66 > PP > PMMA. 

#### 3.2.2. Influence of Plasticizing Pressure

When the plasticizing pressure range was 10–30 MPa, the interfacial friction angle of the PMMA, PP, and PA66 pellets were as shown in Figure 10a–c. The proportion of friction angles of 0–10°, 30–60°, and 80–90° were relatively high. When the plasticizing pressure increased from 10 to 30 MPa, the proportion of interfacial friction angle in the range of 30–60° reduced, while the proportion of others increased accordingly. The influence of plasticizing pressure on the distribution of the interfacial friction angle of pellets were shown in Figure 10d. For PMMA, PP, and PA66, when the plasticizing pressure increased to 30 MPa, the proportion of friction angle of 0–30° were 37.3% (+9.3%), 38.4% (+3.9%), 33.6% (+1.2%); the proportion of friction angle of 30–60° were 17.5% (–18.1%), 23.2% (–14.1%), and 22.1% (−12.4%); the proportion of friction angle of 60–90° were 45.2% (+8.8%), 38.4% (+10.2%), and 44.3% (+11.2%). With increasing the plasticizing pressure, the proportion of interfacial friction angle in the range of 0–30° and 60–90° increased, while the proportion in the range of 30–60° decreased accordingly. Increasing the plasticizing pressure will increase the frictional heat generation efficiency and the compaction degree of the sample, so the plasticizing pressure was negatively correlated with the height of the sample. When the height of the sample decreases, the interfacial friction angle of the pellet tends to be horizontal (0–10°) and vertical (80–90°). From the results, the proportion of interfacial friction angle of pellet in the range of 30–60° was decreased, while the proportion in the range of 0–10°and 80–90° increased accordingly. The end faces of PMMA and PA66 pellets were planar while the side faces were elliptical; the shape of PP was water drop. The influence of the shape of PMMA and PA66 pellets (elliptic cylinder) may be the main cause of why the interfacial friction angle tends to be horizontal (0–10°) and vertical (80–90°) than PP pellets (water drop). In addition, the influence of plasticizing pressure on PMMA and PA66 was greater than that of PP. It takes only 61.53 J for PMMA to rise from room temperature to melting temperature, and 136.01 J for PA66 to rise to the melting point. Thus, the mechanical properties of PMMA decreased more obviously under the same parameters. The influence of plasticizing pressure on the friction angle can be sequenced as PMMA > PA66 > PP. The influence of plasticizing pressure on the distribution of the interfacial friction angle of pellets was more pronounced than the triggering pressure.

#### 3.2.3. Influence of Ultrasonic Amplitude

When the ultrasonic amplitude range was 20–30μm, the interfacial friction angle of the PMMA, PP, and PA66 pellets was as shown in Figure 11a–c. The proportion of friction angles of 0–10° and 80–90° were relatively high. When the ultrasonic amplitude increased from 20 to 30 μm, the proportion of interfacial friction angle in the range of 30–60° reduced significantly, while the proportion of others increased accordingly (about 20%). The influence of ultrasonic amplitude on the distribution of the interfacial friction angle of pellets were as shown in Figure 11d. For PMMA, PP, and PA66, when the ultrasonic amplitude increased to 30 μm, the proportion of friction angle of 0–30° were 36.4% (+5.4%), 39.2% (+11.1%), 35.0% (+1.6%); the proportion of friction angle of 30–60° were 20.2% (−17.7%), 26.2% (−6.2%), and 17.1% (−21.0%); the proportion of friction angle of 60–90° were 43.4% (+12.4%), 34.6% (−5.0%), and 47.9% (+19.3%). With increasing the ultrasonic amplitude, the proportion of interfacial friction angle in the range of 0–30° and 60–90° increased, while the proportion in the range of 30–60° decreased accordingly. Different from the influence of plasticizing pressure, the influence of ultrasonic amplitude on PA66 was greater than PMMA, which may be related to the acoustic impedance of materials. The acoustic impedance reflects the ability of the material to consume sonic energy and was inversely related to the range of ultrasonic propagation. Materials of low acoustic impedance were more affected by ultrasonic [29]. It can be seen from Table 1 that the acoustic impedance of PA66 was 2.9 × 10^5^ Pa s m^−1^, and the acoustic impedance of PMMA was 32.0 × 105 Pa s m^−1^. The influence of ultrasonic amplitude on the friction angle can be sequenced as PA66 > PMMA > PP. 

#### 3.2.4. Influence of Vibration Time

When the vibration time range was 1–2 s, the interfacial friction angle of the PMMA, PP, and PA66 pellets were as shown in Figure 12a–c. The proportion of friction angles of 0–10°, 30–60°, and 80–90° was relatively high. When the vibration time increased from 1 to 2 s, the proportion of interfacial friction angle in the range of 30–60° was reduced, while the proportion of others increased accordingly. The influence of vibration time on the distribution of the interfacial friction angle of pellets was shown in Figure 12d. For PMMA, PP, and PA66, when the vibration time increased to 2 s, the proportion of friction angle of 0–30° were 36.2% (+4.9%), 36.3% (+1.2%), and 39.2% (+6.5%); the proportion of friction angle of 30–60° were 20.0% (−13.7%), 25.1% (−4.7%), 23.8% (−10.2%); the proportion of friction angle of 60–90° were 43.8% (+8.9%), 38.7% (+3.6%), and 36.9% (+3.5%). With increasing the vibration time, the proportion of interfacial friction angle in the range of 0–30° and 60–90° increased, while the proportion in the range of 30–60° decreased accordingly. The influence of ultrasonic amplitude on PMMA was greater than that PA66, and this may be because PMMA was easier to reach the melting point than PA66 under the same process parameters. The influence of vibration time on friction angle can be sequenced as PMMA > PA66 > PP.

### 3.3. Contact Area of the Polymer Pellets

Contact area of the polymer pellets can be classified as inter-pellets, the end face of the cylindrical plasticized sample (end face), and flank face of cylindrical plasticized sample (flank face). For PMMA, PP, and PA66, when the range of trigger pressure was 10–30 MPa, the distribution of the friction surface of the polymer pellets were as shown in Figure 13. During the compression process, polymer pellets not only gradually fill the gap, but also move along the radial direction. Even though the number of friction surfaces of inter-pellets, flank face, and end surface increased, the proportion of the end face decreased due to the higher increased rate of inter-pellets and flank face. When the trigger pressure increased to 30 MPa, the proportion of the inter-pellets were 32.7% (+2.9%), 29.4% (+5.3%), and 42.2% (+14.5%); the proportion of end face were 32.7% (+2.2%), 37.5% (+8.2%), and 28.4% (+1.7%); the proportion of side face were 34.6% (−5.1%), 33.1% (−13.5%), and 29.4% (−16.1%). With increasing the trigger pressure, the proportion of the friction surface (contact area) of inter-pellets and end face increased, while the proportion of the flank face decreased accordingly. The influence of trigger pressure on the friction surface distribution can be sequenced as PA66 > PMMA > PP. 

For PMMA, PP, and PA66, when the plasticizing pressure range was 10–30 MPa, the distribution of the friction surface of the polymer pellets were as shown in Figure 14. With increasing the plasticizing pressure, the heat generation efficiency of friction increased, thus increases the sample density. The results show that the trigger pressure has a greater influence on the friction surface of the inter-pellets than the end face and flank face. When the plasticizing pressure increased to 30 MPa, the proportion of friction surface of inter-pellets were 49.1% (+16.4%), 47.1% (+8.6%), and 47.0% (+10.2%); the proportion of the end face were 33.3% (+5.7%), 27.5% (−6%), and 28.4% (−6.7%); the proportion of flank face were 17.6% (−22.1%), 25.5% (−8.0%), and 24.6% (−3.5%). With increasing the plasticizing pressure, the proportion of the friction surface of inter-pellets increased, while the proportion of the flank face and end face decreased accordingly. The influence of trigger pressure on the friction surface distribution can be sequenced as PMMA > PP > PA66. 

For PMMA, PP, and PA66, when the ultrasonic amplitude range was 20–30 μm, the distribution of the friction surface of the polymer pellets were as shown in Figure 15. When the ultrasonic amplitude increased to 30 μm, the proportion of friction surface of inter-pellets were 46.2% (+14.3%), 49.0% (+10.5%), and 41.9% (+3.8); the proportion of the end face were 27.9% (−9.5%), 28.6% (−4.9%), and 32.3% (−1.3); the proportion of flank face were 25.9% (−4.9%), 22.4% (−5.6%), and 25.8% (−2.5%); With increasing the plasticizing pressure, the proportion of the friction surface of inter-pellets increased, while the proportion of the flank face and end face decreased accordingly. The influence of trigger pressure on the friction surface distribution can be sequenced as PMMA > PP > PA66.

For PMMA, PP, and PA66, when the vibration time range was 1–2 s, the distribution of the friction surface of the polymer pellets were as shown in Figure 16. When the vibration time increased to 2 s, the proportion of the friction surface of inter-pellets changes was 50.5% (+21.0%), 46.4% (+6.8%), and 43.6% (+6.0%); the proportion of end face were 27.7% (−10.7%), 30.5% (−2.4%), and 30.2% (−4.5%) respectively. The proportion of flank face were 21.8% (−10.2%), 23.6% (−3.8%), and 26.2% (−1.5%); With increasing the plasticizing pressure, the proportion of the friction surface of inter-pellets increased, while the proportion of the flank face and end face decreased accordingly. The influence of trigger pressure on the friction surface distribution can be sequenced as PMMA > PP ≈ PA66.

## 4. Conclusions

Ultrasonic plasticizing technology has received extensive attention for its advantages of low energy consumption, good melt fluidity, and short molding time. Friction heat generation is the main heat source in the initial stage of ultrasonic plasticization, which has an important influence on the subsequent viscoelastic heat generation efficiency, determining the temperature field in the whole plasticization process. As a heat generation source strongly affected by the contact interface, the study of frictional heat generation faced many difficulties, such as the random stack of pellets and extremely short duration of the interfacial friction heating. Accurately obtaining the evolution of the interfacial friction angle of polymer pellets and contact area of the plasticizing system is of great significance for establishing a frictional heat generation analysis model that reflects the operating characteristics of ultrasonic plasticizing. In this work, the influence of the process parameter on the interfacial friction angle and contact area of polymer pellets during the initial stage of the ultrasonic plasticization was studied by a super-high magnification lens zoom 3D microscope. The results show that increasing the level of ultrasonic parameters can significantly improve the compression degree of the sample. The plasticized sample is close to the compaction state while the parameter value is high. The proportion of interfacial friction angle of polymer pellets in the range of 0–10°, 80–90°, and 30–60° account for about 16%, 18%, and 36% while the parameter level is low. With increasing parameter level, the proportion of interfacial friction angle in the range of 0–10° and 80–90° increased, while the proportion in the range of 30–60° decreased accordingly, about 20%, 24%, and 24%. The proportion of the contact area of the polymer pellets increased up to 50% interfacial friction area which includes the upper, lower, and side area of the cylindrical plasticized sample. The influence of the triggering pressure on the friction state of the plasticized samples is similar to the ultrasonic parameter but to a relatively low degree. 

## Figures and Tables

**Figure 1 polymers-11-02103-f001:**
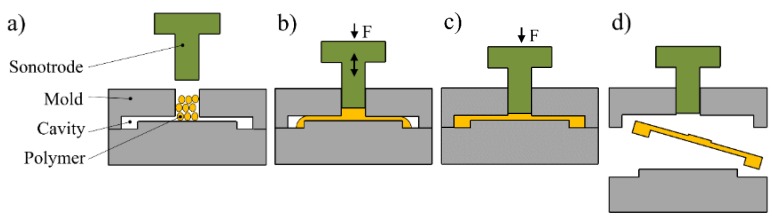
Process diagram of ultrasonic plasticization micro-injection molding: (**a**) Feeding; (**b**) plasticizing and injection; (**c**) holding; (**d**) take out parts.

**Figure 2 polymers-11-02103-f002:**
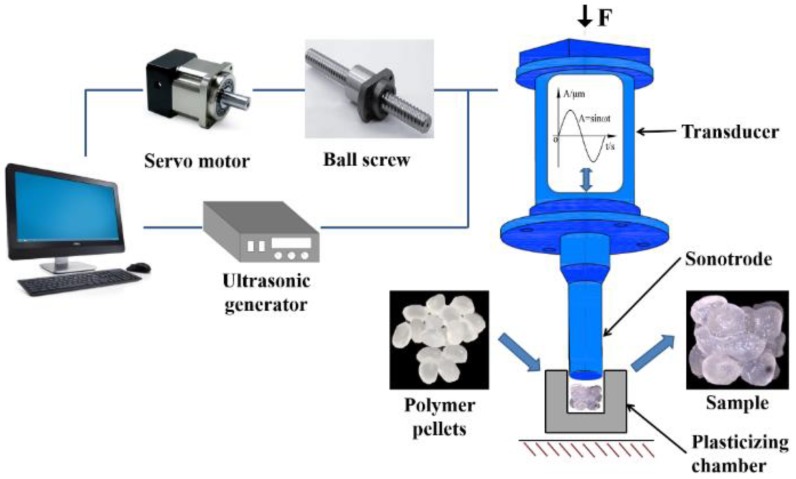
Working principle of the UPMIM machine.

**Figure 3 polymers-11-02103-f003:**
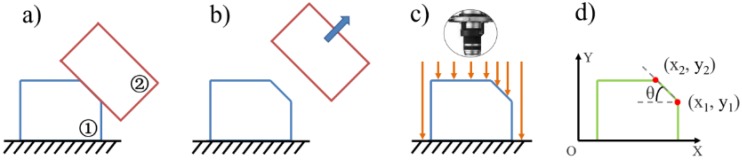
Schematic diagram of the measuring method for the interfacial friction angle of pellets: (**a**) Select the friction angle to be measured; (**b**) remove the pellet; (**c**) measuring; (**d**) statistical analysis.

**Figure 4 polymers-11-02103-f004:**
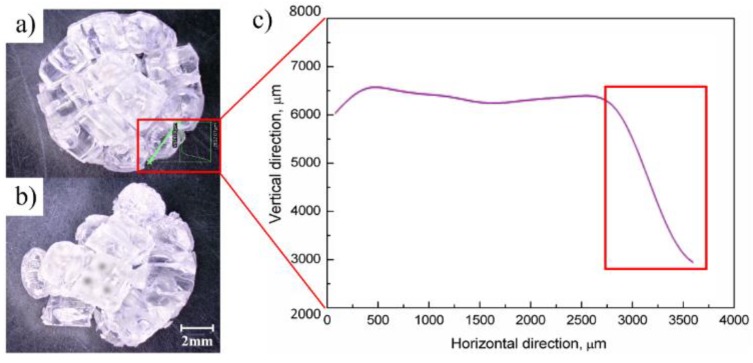
Process and result of friction angle measurement: (**a**) Measuring friction angle; (**b**) Remove measured pellets; (**c**) Results of measurement.

**Figure 5 polymers-11-02103-f005:**
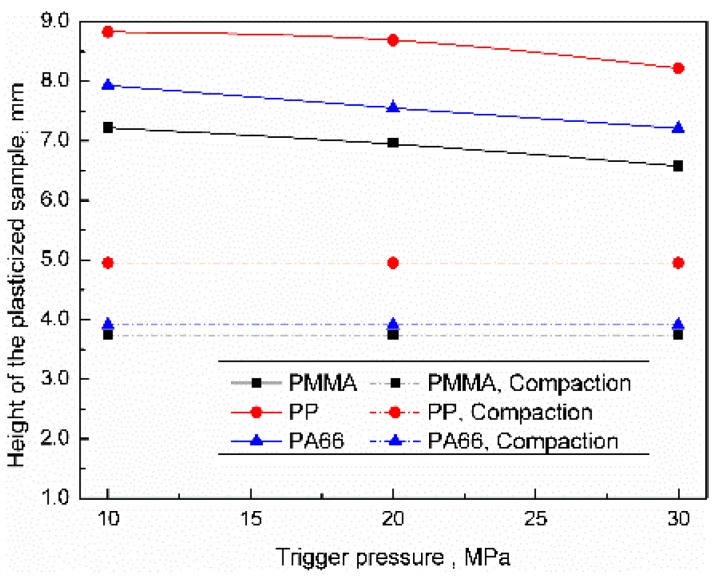
Influence of trigger pressure on the height of the plasticized sample.

**Figure 6 polymers-11-02103-f006:**
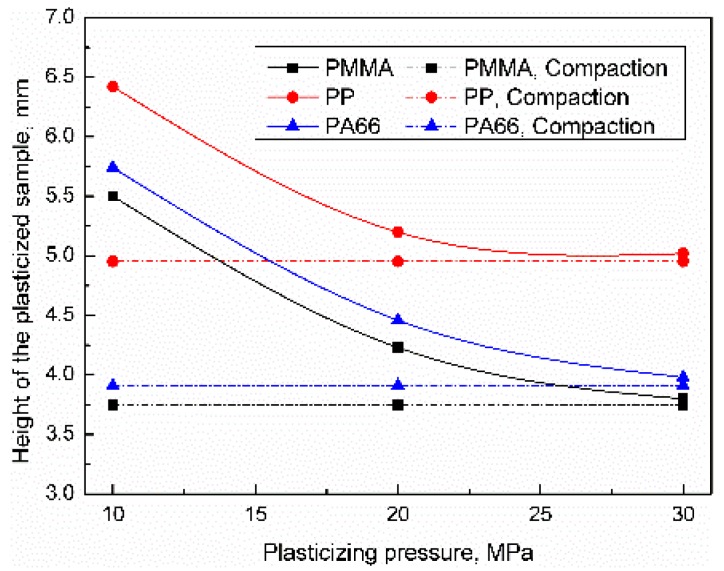
Influence of plasticizing pressure on the height of the plasticized sample (UA = 25 μm, VT = 1.5 s).

**Figure 7 polymers-11-02103-f007:**
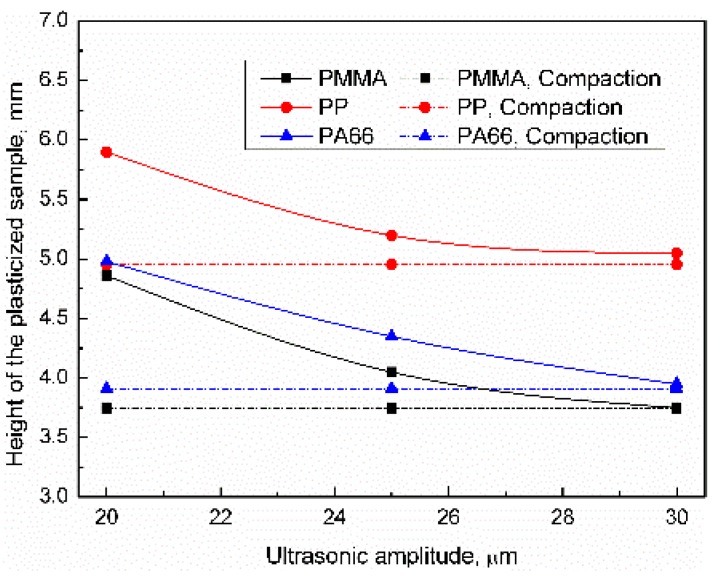
Influence of ultrasonic amplitude on the height of the plasticized sample (UT = 1.5 s, PPe = 20 MPa).

**Figure 8 polymers-11-02103-f008:**
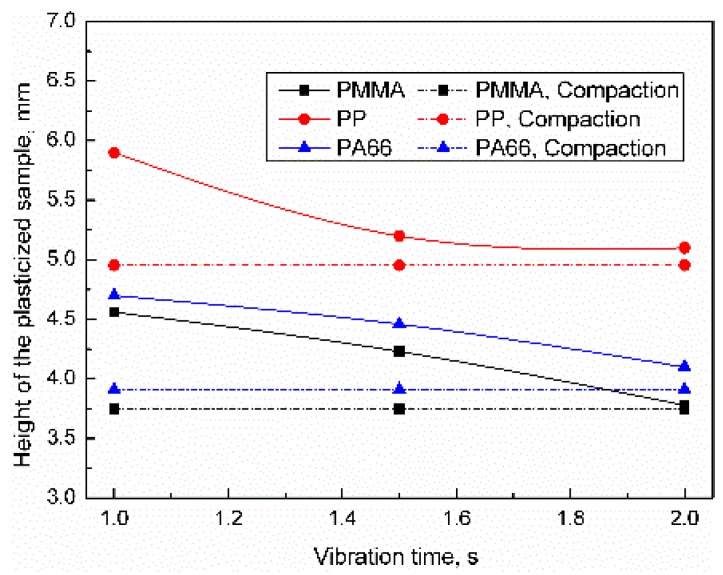
Influence of vibration time on the height of the plasticized sample (UA = 25 μm, PPe = 20 MPa).

**Figure 9 polymers-11-02103-f009:**
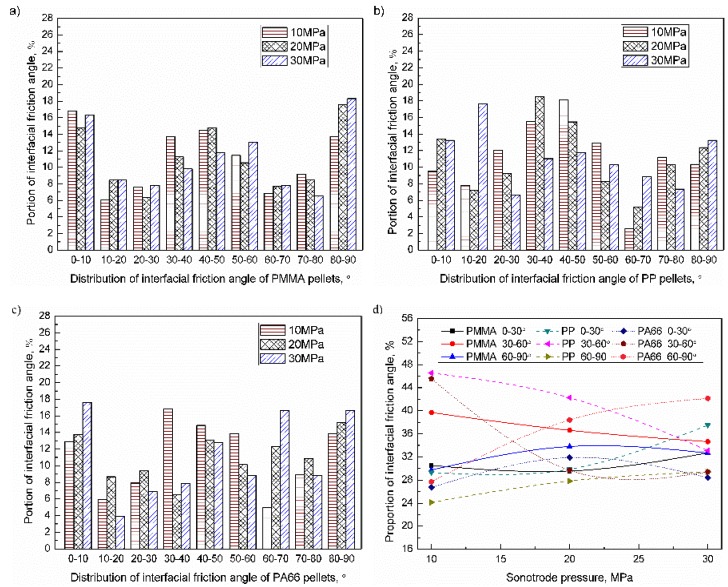
Influence of trigger pressure on the distribution of the inter-pellets friction angle of pellets: (**a**) PMMA; (**b**) PP; (**c**) PA66; (**d**) Evolution of the inter-pellets friction angle.

**Figure 10 polymers-11-02103-f010:**
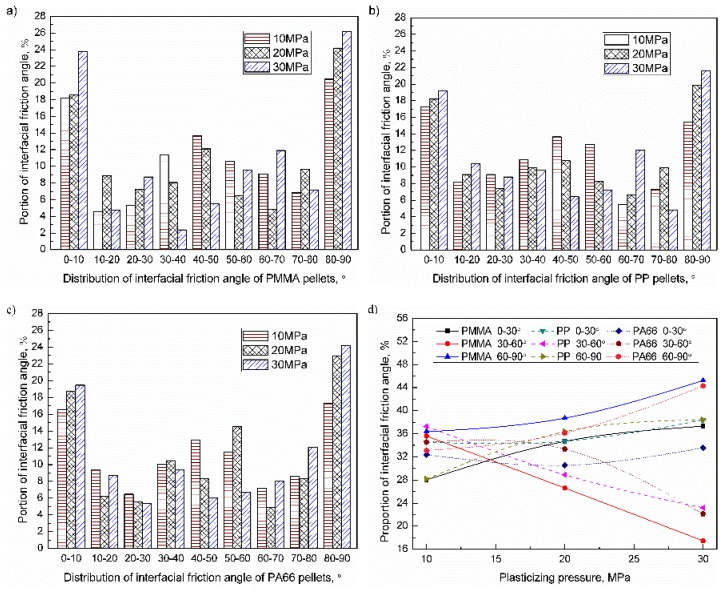
Influence of plasticized pressure on the distribution of inter-pellets friction angle of plasticized samples (UA = 25 μm, VT = 1.5 s): (**a**) PMMA; (**b**) PP; (**c**) PA66; (**d**) Evolution of the inter-pellets friction angle.

**Figure 11 polymers-11-02103-f011:**
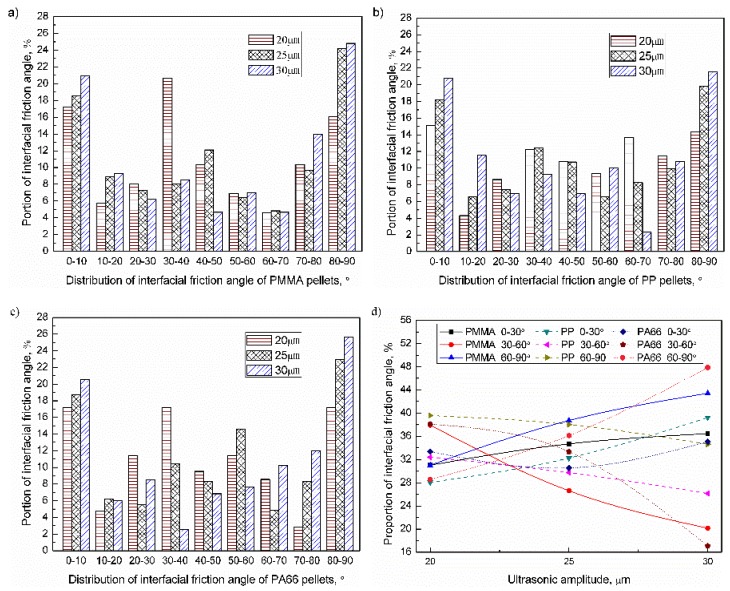
Influence of ultrasonic amplitude on the distribution of inter-pellets friction angle of plasticized samples (VT = 1.5 s, PPe = 20 MPa): (**a**) PMMA; (**b**) PP; (**c**) PA66; (**d**) Evolution of the inter-pellets friction angle.

**Figure 12 polymers-11-02103-f012:**
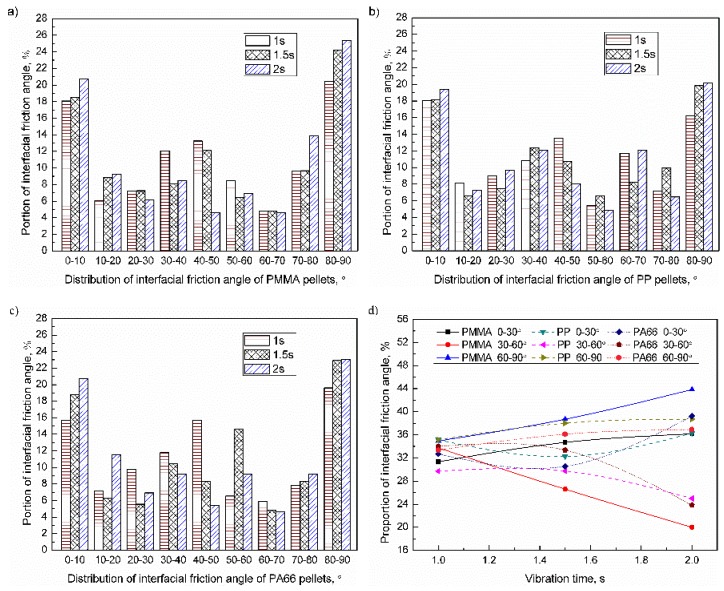
Influence of vibration time on the distribution of inter-pellets friction angle of plasticized samples (UA = 25 μm, PPe = 20 MPa): (**a**) PMMA; (**b**) PP; (**c**) PA66; (**d**) Evolution of the inter-pellets friction angle.

**Figure 13 polymers-11-02103-f013:**
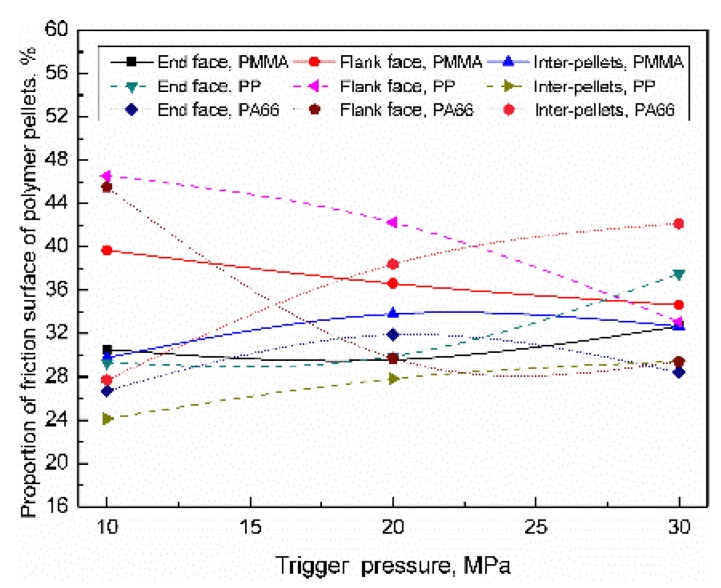
Influence of trigger pressure on the distribution of the contact area of the polymer pellets.

**Figure 14 polymers-11-02103-f014:**
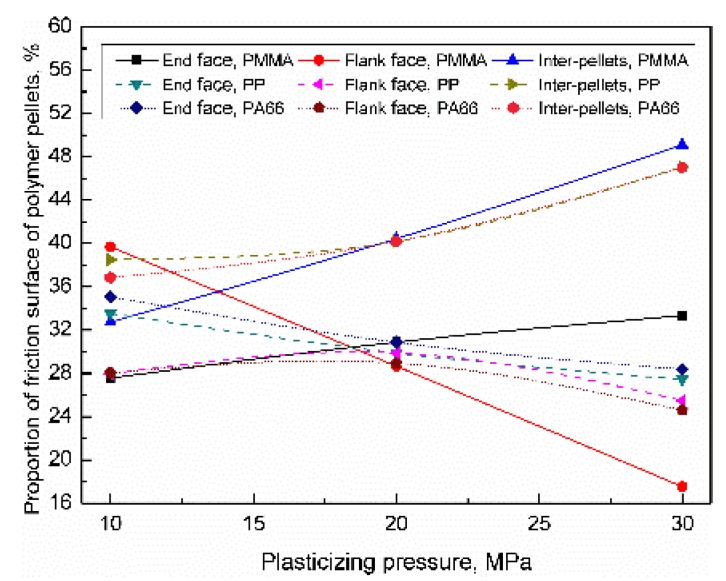
Influence of plasticizing pressure on the distribution of the contact area of the polymer pellets (VT = 1.5 s, PPe = 20 MPa).

**Figure 15 polymers-11-02103-f015:**
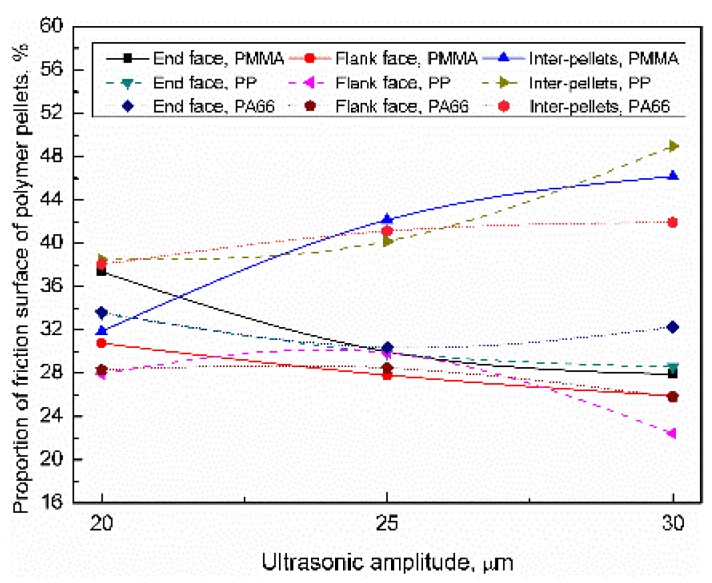
Influence of ultrasonic amplitude on the distribution of the contact area of the polymer pellets (VT = 1.5 s, PPe = 20 MPa).

**Figure 16 polymers-11-02103-f016:**
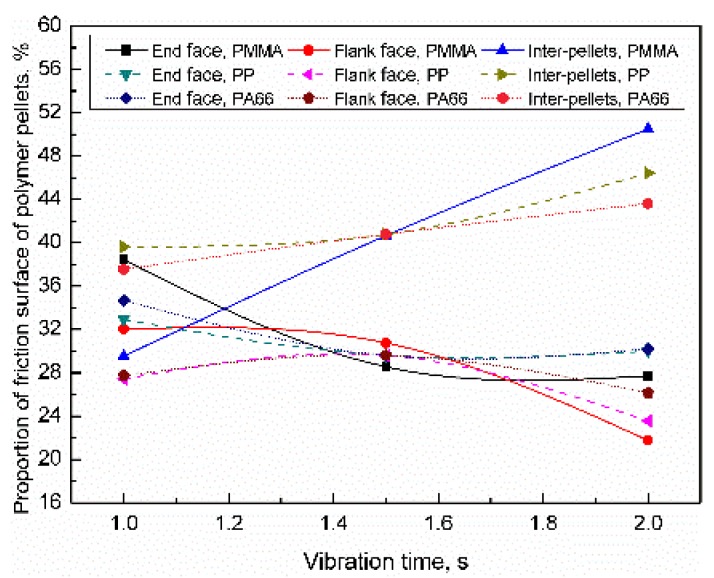
Influence of vibration time on the distribution of the contact area of the polymer pellets (VT = 1.5 s, PPe = 20 MPa).

**Table 1 polymers-11-02103-t001:** Material properties.

Material	Crystal Type	Density (g/cm^3^)	Melt Point (°C)	Specific Heat Capacity (KJ/Kg·°C)	Elastic Modulus (Gpa)	Acoustic Impedance (10^5^ Pa·s∙m^−1^)
PMMA	Amorphous	1.19	130~140	1.465	3.3	32.00
PP	Semi-crystalline	0.91	175	1.926	0.9	11.10
PA66	Crystalline	1.14	252	1.675	2.6	2.90

**Table 2 polymers-11-02103-t002:** Parameter combination of single factor experimental.

Number	Ultrasonic Amplitude (UA)/μm	Plasticization Pressure (PPe)/MPa	Vibration Time (VT)/s
1	20, 25, 30	20	1.5
2	25	10, 20, 30	1.5
3	25	20	1, 1.5, 2

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
