# Peer review of "Evolution of Interfacial Friction Angle and Contact Area of Polymer Pellets during the Initial Stage of Ultrasonic Plasticization"

_polymers, 2019, doi:10.3390/polym11122103_

Round 1

Reviewer 1 Report

In this article, the authors investigated the effects of trigger pressure, plasticizing pressure, ultrasonic amplitude, and vibration time on the interfacial friction angle and the contact area of the polumer pellets, by using PMMA, PP, and PA66 as polymer samples. The authors did a great job on investigating each parameter, and the work has been done thoroughly. However, the authors need to address the following concerns:

In section 3.1, the authors mentioned the temperature of the contact area of the pellet increases rapidly under interfacial friction (line 182). Can the authors prove this point? The writing of the unit is not consistent. For example, the authors sometimes wrote Mpa but sometimes wrote MPa. The time terms are not consistent as well. When describing the results, the past term should be used. However, the authors often described by using present term. In section 3.2, the authors wrote out the experimental data in details. It might be better to summarize in tables, thus the readers will not be lost in details without understanding the main concepts. In section Results and Discussion, the portion of discussion is not enough. The authors mainly described the results, but did not explain the potential mechanism or did not compare with other researchers’ work. The authors also did not discuss the strengths and weaknesses of their work. The authors mentioned the difficulties of the current method: the random stack of pellets and extremely short duration of the interfacial friction heating. Do the authors consider their own method solve the difficulties of the current method? The authors should discuss on this point more in order to make readers understand the unique characteristics of this present work.

Author Response

Response to Reviewer 1 Comments

Point 1: In section 3.1, the authors mentioned the temperature of the contact area of the pellet increases rapidly under interfacial friction (line 182). Can the authors prove this point?

 Response 1: In previous research on the heat generation mechanism of ultrasonic plasticization, we found that the temperature of the friction interface will rise from room temperature to viscous flow temperature within 2s, related works were as follows:

Tao P, Bingyan J, Yang Z. Study on the Mechanism of Interfacial Friction Heating in Polymer Ultrasonic Plasticization Injection Molding Process[J]. Polymers 2019, 11(9), 1407. Bingyan J, Huajian P, Wangqing W, et al. Numerical Simulation and Experimental Investigation of the Viscoelastic Heating Mechanism in Ultrasonic Plasticizing of Amorphous Polymers for Micro Injection Molding[J]. Polymers, 2016, 8(5):199. Wu W, Peng H, Jia Y, et al. Characteristics and mechanisms of polymer interfacial friction heating in ultrasonic plasticization for micro injection molding[J]. Microsystem Technologies, 2017, 23(5):1385-1392.

The references above have been inserted in section 3.1(line 204).

 Point 2: The writing of the unit is not consistent. For example, the authors sometimes wrote Mpa but sometimes wrote MPa. The time terms are not consistent as well. When describing the results, the past term should be used. However, the authors often described by using present term.

Response 2: Units (text, figure, and table) and tenses have been checked and corrected.

Point 3: In section 3.2, the authors wrote out the experimental data in details. It might be better to summarize in tables, thus the readers will not be lost in details without understanding the main concepts. In section Results and Discussion, the portion of discussion is not enough. The authors mainly described the results, but did not explain the potential mechanism or did not compare with other researchers’ work.

Response 3: The structure of Results and Discussion have been streamlined and adjusted (such as line297); the discussion section of the results was more prominent and clear. Without changing the meaning, the number of words has been reduced from 7207 to 6046.

Point 4: The authors did not discuss the strengths and weaknesses of their work. The authors mentioned the difficulties of the current method: the random stack of pellets and extremely short duration of the interfacial friction heating. Do the authors consider their own method solve the difficulties of the current method? The authors should discuss on this point more in order to make readers understand the unique characteristics of this present work.

Response 4: In previous works, researchers in this field established simplified physical models referring to ultrasonic welding. We hold that it was inadequate to establish an accurate model without considering the contact characteristic during ultrasonic plasticization process, but no relevant research in this field was found. It was difficult to acquire interfacial friction angle in real-time due to the random stack of pellets and extremely short duration of the interfacial friction heating. In this work, by studying with plasticized samples obtained at various time intervals, the evolution of the interfacial friction angle and contact area of polymer pellets during the ultrasonic plasticization was firstly acquired by super-high magnification lens zoom 3D microscope. The results can be used to establish an accurate geometric analysis model of friction heat generation.

Related background and unique characteristics description of this work have inserted in Introduction (line 69).

Reviewer 2 Report

Please see the following suggestions:

1 - Improve the text writing in a more concise way.

2 - Page 2 - line 47 - whre is "...technology is suffered..." write "...technology is subjected..."

Page 5- line 164/165 - where is "The main reason for the height of the sample much higher than ...may be polymer pellets were unplasticized..." write "The main reason for a height of the sample much higher than ...may be that polymer pellets were unplasticized...

3 - Replace Mpa by MPa wherever this occurs

Author Response

Response to Reviewer 2 Comments

Point 1: Improve the text writing in a more concise way.

Response 1: The structure of Results and Discussion have been streamlined and adjusted (such as line297); the discussion section of the results was more prominent and clear. Without changing the meaning, the number of words have been reduced from 7207 to 6046.

Point 2: Page 2 - line 47 - where is "...technology is suffered..." write "...technology is subjected..."

Page 5- line 164/165 - where is "The main reason for the height of the sample much higher than ...may be polymer pellets were unplasticized..." write "The main reason for a height of the sample much higher than ...may be that polymer pellets were unplasticized...

Response 2: The sentence have been checked and corrected.

Point 3: Replace Mpa by MPa wherever this occurs

Response 3: Units (text, figure, and table) and tenses have been checked and corrected.